# Eurasian back-migration into Northeast Africa was a complex and multifaceted process

Rickard Hammarén[1,2], Steven T. Goldstein[2], Carina M. Schlebusch[1,3,4]*

**1** Human Evolution, Department of Organismal Biology, Evolutionary Biology Centre, Uppsala University, Uppsala, Sweden, **2** Department of Archaeology, Max Planck Institute for the Science of Human History, Jena, Germany, **3** Palaeo-Research Institute, University of Johannesburg, Johannesburg, South Africa, **4** SciLifeLab, Uppsala, Sweden

* carina.schlebusch@ebc.uu.se

**Data Availability Statement:** All data used in the study is from published studies. For some datasets permission from the original authors were obtained through data access agreements.

## Abstract

Recent studies have identified Northeast Africa as an important area for human movements during the Holocene. Eurasian populations have moved back into Northeastern Africa and contributed to the genetic composition of its people. By gathering the largest reference dataset to date of Northeast, North, and East African as well as Middle Eastern populations, we give new depth to our knowledge of Northeast African demographic history. By employing local ancestry methods, we isolated the Non-African parts of modern-day Northeast African genomes and identified the best putative source populations. Egyptians and Sudanese Copts bore most similarities to Levantine populations whilst other populations in the region generally had predominantly genetic contributions from the Arabian peninsula rather than Levantine populations for their Non-African genetic component. We also date admixture events and investigated which factors influenced the date of admixture and find that major linguistic families were associated with the date of Eurasian admixture. Taken as a whole we detect complex patterns of admixture and diverse origins of Eurasian admixture in Northeast African populations of today.

## Introduction

Northeast Africa has undeniably been a key region in human evolutionary history. The out-of-Africa migrations need to have passed through, if not originated in the region. East Africa is also home to some of the most important fossil evidence for human evolution from early bipedal species such as Australopithecus afarensis [1], to the emergence of early anatomically modern humans such as the Omo fossils [2, 3].

Numerous overlapping migrations of farmers and herders over the last several thousand years have also played a critical role in reshaping the current socioeconomic and linguistic diversity of the region. It is clear that back migrations into Northeast Africa have had a major impact on the genetic ancestries of the peoples in the region today [4–6]. Ethiopian populations, for instance, harbor a large proportion of "non-African" ancestry, as high as, ∼ 40% in some groups—see for instance the Amhara in Fig 1c in [7]. What is clear is that some current-day Northeast Africans can trace much of their ancestry from other sources than the original

**Funding:** The project was funded by the European Research Council (ERC StG AfricanNeo, grant no. 759933) and the Knut and Alice Wallenberg Fellowship grant. Computations were enabled by resources provided by the Swedish National Infrastructure for Computing (SNIC) at UPPMAX, partially funded by the Swedish Research Council (through grant agreement no. 2018-05973). The funders had no role in study design, data collection and analysis, decision to publish, or preparation of the manuscript. The salaries of C.S. and R.H. was funded by the European Research Council (ERC StG AfricanNeo, grant no. 759933) and the Knut and Alice Wallenberg Fellowship grant.

**Competing interests:** The authors have declared that no competing interests exist.

hunter-gatherers in the region, such as the Mota individual, an Ethiopian male who lived around 4500 years ago [8, 9]. It is also clear that these back migrations into Africa have been ongoing for a long time period. For North Africa, seven individuals from Morocco that had a high affinity to Middle Eastern populations, dated to 15 000 years ago, suggesting the possibility that similar deep-in-time admixtures might have occurred in other parts of Africa [10].

In 2017, several ancient genomes were sequenced in an attempt to uncover the demographic patterns in African prehistory [11]. The study contained data from 16 ancient African individuals from 8 100—400 BP. They found that ancient East African hunter-gatherers form a cline of ancestry with modern-day southern African hunter-gatherer (San) groups. This indicates that in the past, hunter-gatherers with a gradient of shared ancestry ranged from eastern to southern Africa. The fact that these hunter-gatherer groups existed until the relatively recent past allows for the possibility of interactions between them and later pastoralist and agricultural groups in East Africa. This data was later re-analyzed with several new individuals, particularly from East Africa [5]. They proposed a four-stage model where initially Sudanese Nilotic speakers admixed with groups with Eurasian ancestry (either from Northern Africa or the Levant) within Northeast Africa. In step two, the descendants of these groups migrated to East Africa reaching Lake Turkana by around 5 000—4 000 BP and central Tanzania by around 3000 BP and mixed with local hunter-gatherer groups throughout this process [5]. The first signs of pastoralism in East Africa coincide with these events. Thirdly the second wave of Sudanese-related groups migrated into the area and contributed to the pastoral Iron Age populations. Lastly, West African ancestry (genetically similar to Bantu speakers) appeared alongside the advent of crop farming in the region. These findings were then yet again revised in 2020 [12]. By analyzing 20 additional ancient individuals, additional resolution was given to the picture and several new patterns emerged. Mainly, [12] propose that the pastoralists probably arrived in East Africa in multiple waves from several different locations, or that severe population structure was present (distinguishing between the two was not possible). Both [12] and [5] conclude that there was no single event of hunter-gatherer and herder introgression, neither in space nor in time. Instead a complex "moving frontier" is proposed with diverse patterns of interactions along the contact zones between hunter-gatherer and herder groups.

In the last decade, several genetic studies on modern-day populations have focused on the genetic demographic history of Ethiopia and found patterns of linguistic stratification within Ethiopian populations, i.e. populations within the same language family are more similar to each other than populations belonging to other language families [4, 13, 14]. It is less clear if this pattern holds true in Northeast Africa as a whole, as [15] found a stronger association between geography and genetics than between genetics and linguistic family. By studying modern-day genetic variation, [6] investigated the non-African part of Ethiopian populations and were able to conclude that there has been Eurasian admixture, likely coming from Levantine, rather than Arabian populations. This event was estimated to have occurred around 3 000 years ago.

By leveraging one of the largest datasets of Northeast African populations to date, we aim to add resolution to Eurasian admixture in Northeast African populations. Specifically, we aim to improve the estimation of the best proxies for the origin of Eurasian admixture in modern-day Northeast African populations by using more Northeast African and Middle Eastern, and Eurasian reference populations. In this study, we follow the approach of [4, 6, 16] in that we employ local ancestry methods to identify the Eurasian fragments of East African genomes and extract those segments from the surrounding genomes, a process referred to here as ancestry-deconvolution. We then identify the current-day populations that best match those segments. We also date the events to get a better understanding of historic and prehistoric movements in the region. Using the information of possible sources for admixture and dating

of these, we construct a model representative of the population history in the region. Overall we find a complex history of Eurasian admixture in Northeastern Africa, related to the spread of languages, the Muslim conquest, and trade routes along the Red Sea.

## Materials and methods

### Genotyping QC

Genotyping data was gathered from previously published studies [4, 7, 17–26]. See S6 Table for a list of populations included in this study, their original population, language classification, and geographic coordinates. The geographic sampling information is displayed in S1 Fig. Only autosomal chromosomes were investigated in this study. PLINK v1.90b4.9 [27] was used for data handling and processing. Before data merging, each dataset was quality controlled which entailed 1) removing related individuals using KING [28], the first individual within each pair of second-degree relatives or closer was removed 2) SNPs with less than 1% genotyping rate was excluded (`plink --geno 0.01`) 3) C/G and A/T SNPs were removed 4) individuals with at least 15% missingness was removed (`plink –mind 0.15`) 5) potential genotyping errors were removed (`plink --hwe 0.0000001`) 6) lastly only overlapping SNPs between the datasets were kept.

Before analysis that could be adversely affected by linkage disequilibrium (ADMIXTURE and PCA) SNPs in LD were filtered out using `plink --indep-pairwise 50 10 0.1`.

The data from [4] was converted from hg18 to hg19 using the `liftOver` tool from UCSC (https://genome.ucsc.edu/cgi-bin/hgLiftOver).

As the number of individuals in each population varied quite substantially, from only a few individuals to around a hundred for other populations, we randomly sub-sampled all populations down to 30 individuals. This was done to reduce the effect that sample size can have on demographic inference.

### Metadata

Geographic information about the populations was gathered from the original publications in the following fashion, 1) directly from the text or 2) if no coordinates were provided then they were interfered from the map of sampling locations, 3) if no map or coordinates were supplied, then a point in the middle of the respective country was chosen this was the case for three publications [17, 19, 25]. Regarding language classification, we followed a similar approach as for geographic data, namely that information/classification was used if available in the original publication. The Egyptians from [25] and the Qatari from [19] were both assumed to be Arabic speakers and thus classified as Semitic. The Niger-Kordofanian classification used in [7] was changed to Niger-Congo, to better align with the other datasets. For visualization purposes, the Semitic speakers on the African continent were given their own label (African Semitic) and their own colour. This distinction was only made to better distinguish between the investigated populations (targets) and Middle Eastern populations used as references. This distinction is thus based solely on geography and is not supported by any linguistic deductions. For a detailed classification of all linguistic groupings used, see S6 Table.

### Population structure inferences

Unsupervised population structure inference analysis for K = 2 to K = 15 was performed with ADMIXTURE [29] version 1.3.0 for K = 2 to K = 15 using a random seed each time and repeated 50 times. PONG version 1.5 [30] was used to visualize the results and find the major mode and pairwise similarity within the major modes. Principal component analysis (PCA)

was performed using `FlashPCA` version 2.0 [31]. For the PCA plots, PC refers to Principal Component, with each value in the PCA plots representing the projection of the data on the eigenvectors, scaled by the eigenvalues. Uniform Manifold Approximation and Projection for Dimension Reduction (UMAP) was performed on the genotypes directly using the `umap-learn` python library version 0.5.2.

### Patterns of migration rates

The migration rate over the sampling area was investigated using FEEMS [32]. A grid of coordinates covering North-Eastern Africa and most of the Middle East was generated. Cross-validation was performed and the lambda with the lowest cross-validation value was used to generate the final plot.

### Phasing

Phasing was carried out out using `SHAPEIT` version 2.r837 [33] with the 1000 genomes phase 3 reference genomes [18] and options `--states 500 --main 20 --burn 10 --prune 10`. Misaligned sites between the reference dataset and panel were excluded.

### Local ancestry estimation

`MOSAIC` version 1.3.7 was compiled and ran under `R` version 4.0.0 [34] to perform local admixture inference, admixture dating as well as ancestry deconvolution. To minimize the potential bias of different sample sizes between investigated target populations, and sources the number of individuals investigated for each population was downsampled to ten individuals. The ancestry deconvolution was performed by running MOSAIC, using the specified resources (see Results for specific scenarios), and then looking at the constructed ancestries that MOSAIC infers from the provided sources. The constructed ancestry in MOSAIC was then compared to the source populations and $F_{st}$ was used to evaluate which one of the source ancestries it most closely resembled. If one of the ancestries shows the most genetic similarity to a Eurasian source then the analysis continued for that ancestry. Thus only samples/targets that mosaic found could be explained by at least one Eurasian ancestry source was ancestry deconvoluted. Segments of each individual's genome that were assigned to the Eurasian ancestry with a probability of 80% or more by MOSAIC were saved and the remainder of the genome was set as missing. Admixture dating was extracted from MOSAIC's co-ancestry curves for the Eurasian-like ancestry.

### Outgroup $f_3$

Outgroup $f_3$ were performed using `qp3Pop` from `ADMIXTOOLS 2` version 2.0 [35, 36]. The San population Ju|'hoansi was used as the outgroup and the extracted ancestry fragments of each target population were tested against populations from the Eurasian reference dataset. The aim of this procedure is to identify which reference population is most like the extracted Eurasian ancestry.

### Visualization

PCA and outgroup $f_3$ results were visualized in R version 3.6.1 using the libraries `ggplot2` [37]. Maps were created in R version 3.6.1 using `ggplot2` and the `sf`, `rnaturalearth`, and `rnaturalearthdata` libraries, the latter being based on the public domain maps and rasters from Natural Earth @ naturalearthdata.com. The kriging projection maps were generated in Surfer version 12.0.626 from Golden Software.

# Results

After quality control, removal of related individuals, and down-sampling to a maximum of 30 individuals per population the dataset consisted of 2066 individuals from 101 population groups and 199 422 SNPs. Note that some populations are represented multiple times from different original publications, resulting in a total of 97 unique populations, see S6 Table.

## Population structure in Northeast Africa

General population structure inferences were performed using PCA and ADMIXTURE on a dataset where SNPs in LD were pruned (85 529 SNPs remaining). The output from ADMIXTURE shown in Fig 1 (for full analysis see S2 Fig) captures similar patterns to the PCA analysis (S5–S19 Figs), with the first separations being between major geographic regions. The K with the lowest cross-validation error was K = 13, see S3 Fig.

The first division in the data is between Africans and non-Africans, and it is clear that North- and East- Africans have a much larger proportion of shared ancestry with Eurasian groups than other African groups (K = 3 in S2 Fig). East African groups break away from other African groups at K = 5 via a component (black) maximized in the Nuba at 80.1%. Of particular interest for the present investigation is also the component that emerges is K = 8 (light orange) maximized in the Ari, Sabue and Gumuz populations. The Sabue is one of the few remaining hunter-gatherer groups in East Africa today and they share genetic continuity with earlier hunter-gatherer ancestry from the region [38] represented by the Mota individual [8]. The Ari, Gumuz and Sabue have been suggested to retain a high degree of ancient East African hunter-gatherer ancestry, [4, 26, 38] and our demographic analyses indicate a high degree of similarity between these populations. This component is shared with many other East African groups, displaying fractions of ancestry that show deep continuity in the region.

At K = 11 another East African component appears, maximized in the Somali populations and might represent Cushitic-related ancestry. Levantine populations separate from the Arabic populations at K = 14 and we visualized these two components using a Kriging interpolation across the study area, Fig 1. These two ancestries were the component maximized in the Lebanese Druze (dark blue) and the component highest in the Yemeni (blush pink).

To investigate the differences in affinities of our target populations to either Levantine and Yemenite ancestry we performed a $f_4$ test. The test took the following form Yemen_YEMEN | Lebanese_Christian|Target S22 Fig. It showed significant association with Levantine for the populations north of the Sudanese BeniAmer as well as for the Oromo and Tygray from Ethiopia, the Kenyan Luhya and the Maasai from Kenya. No populations had a significantly higher association to Yemenite ancestry when compared to Levantine ancestry in this more stringent test. The test however highlight the importance of Levantine admixture for more northern populations in particular.

In our Principal Component (PC) analyses, the first PC differentiates between African and non-African groups S5 Fig. Several African populations fall on the cline between African and non-African variation, in particular North Africans, such as the Egyptians and populations from Sudan who are known to have Eurasian admixture [15]. We also observe a grouping according to linguistics where Omotic, Afro-Asiatic, and Nilo-Saharan speakers separate from each other. East African groups are positioned on the diagonal between PC 1 and PC 3, with the Ari, Sandawe, and Sabue populations forming their own cluster in the direction of the southern African Khoe-San, indicating shared ancestries between these hunter-gatherer groups S5 Fig. This cline is similar to what was found in studies using aDNA [5, 11] and is a reflection of the cline between southern and East African hunter-gather ancestry.

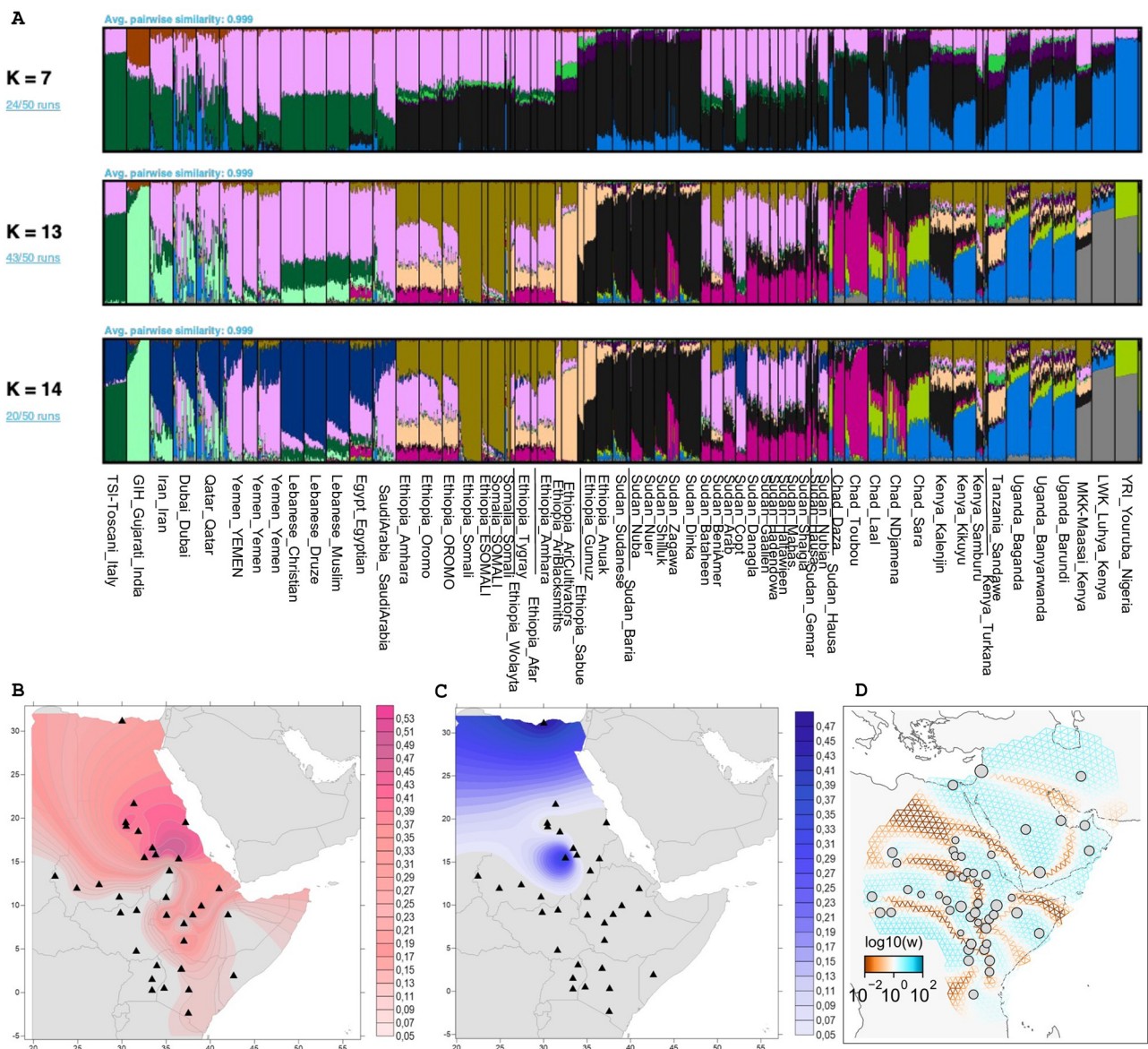

**Fig 1. Spatial distribution of ancestries.** A) ADMIXTURE results for K = 7, 13 & 14 visualized using PONG (a truncated version of S2 Fig, and the K = 13 and 14 panels do not include all clusters, e.g. East Asia is not represented). B) Kriging plot of the distribution of the pink component at K = 14 (maximized in Yemen) in A on the East Africa populations. C) Kriging plot of the distribution of the dark blue component (Lebanese) from K = 14. D) FEEMS plot of inferred patterns of migration rates for the lowest cross-validation lambda. High w (blue) indicates an area with higher than average effective migration whilst low w (brown) indicates lower than average effective migration areas.

UMAP was also performed in the genotype information in our dataset, see S20 Fig. This analysis produces two larger clusters of populations, one consisting of West African groups, Eastern Bantu speakers, the Saharan speakers, the Nuer, Dinka, and Shiluk from Sudan. The other cluster contains mainly Middle Eastern populations and Ethiopians, as well as the remaining Sudanese populations.

To further investigate the historic patterns of gene flow, migrations and which barriers to migrations are evident across the region of interest, we used the FEEMS software package [32], Fig 1D.

## Determination of Eurasian sources through local ancestry estimation

To identify distinct ancestries in East African populations, we employed MOSAIC [34]. We wanted to identify patterns of local ancestries and determine which of our reference populations were the best proxies for the different genetic components. In particular, we were interested in the "non-African" or rather Eurasian segments of the genomes. Following the approach from previous studies, we aimed to isolate these Eurasian genetic segments and analyze them in isolation [6, 39]. Thirty-five East African and Northeast African populations were chosen as target populations to analyze. For location and linguistic groups of these target populations see S4 Fig.

As has been shown in previous studies, and indicated by our demographic inference Fig 1 and S6 Fig, there are generally four main components of modern-day East African genetic ancestry [5, 12]. Namely, basal East African hunter-gatherer ancestry, Sudanese/Nilotic ancestry, Eurasian ancestry, and West African ancestry associated with Bantu speakers. Since the aim of this study is to identify the best proxy for the source of the Eurasian ancestry of the Northeast African populations, we constructed a scenario where we could use these four ancestral sources to paint the haplotypes of our chosen target populations using MOSAIC [34]. We set up an initial scenario to try and identify the best Eurasian source to use for further analyses. In this scenario we used the following populations: To represent the basal East African hunter-gatherer ancestry we chose the Sabue [26, 38]. The Sabue has been referred to by many different names in the literature, for instance, Shabo and Chabu, here we use the name used in the original publication of the data [26]. The CEU (Utah residents with Northern and Western European ancestry) population from the 1000 genomes consortium was chosen as a proxy for general Eurasian ancestry. The Sudanese Dinka was chosen to represent Sudanese ancestry (the group that defines the black component in the ADMIXTURE analysis associated with Sudanese ancestry). The YRI (Yoruba in Ibadan, Nigeria) was used as a proxy for West African Niger-Congo and Bantu-speaker-associated ancestry.

We then used these four populations (CEU, YRI, Dinka, and Sabue) as sources in a 3-way admixture scenario in MOSAIC and extracted the called genotypes that were assigned to the CEU-like ancestry with a probability of 80% or more. The closest affinity of each constructed ancestry was determined by the $F_{st}$ test in MOSAIC against the four source populations. This resulted in regions of each East African individual's genome that is more closely associated with a Eurasian ancestry than with the other ancestries. Only these regions were kept whilst the rest was set as missing for each individual see S5 Table for missingness information of each population.

This non-African part of the genomes was then compared using outgroup $f_3$ to Eurasian references populations with the Ju|'hoansi as outgroup (target |REF |Ju|'hoansi). A higher value of the outgroup $f_3$ test indicates a smaller genetic distance between the in-groups compared to the outgroup. The San group Ju|'hoansi was chosen as the outgroup since previous studies had shown them to be the least admixed modern-day Khoe-San group [23]. The $f_3$ outgroup test thus identified the population that is the most similar to the Eurasian fraction of the Northeast African target populations, see the top three in Fig 2 and top five in S2 Table.

The outgroup $f_3$ analysis provided us with the best Eurasian source population to use for each of our Northeast African target populations. We then re-ran the MOSAIC analysis above but instead used this best source instead of the CEU. We refer to this dataset and approach as the "best by $f_3$". This approach can also be thought of as using our prior knowledge to construct the best scenario.

As an alternative to our own ascertained approach we also tested other less constrained scenarios. In these scenarios, we kept YRI, Sabue, and Dinka as the three African populations and

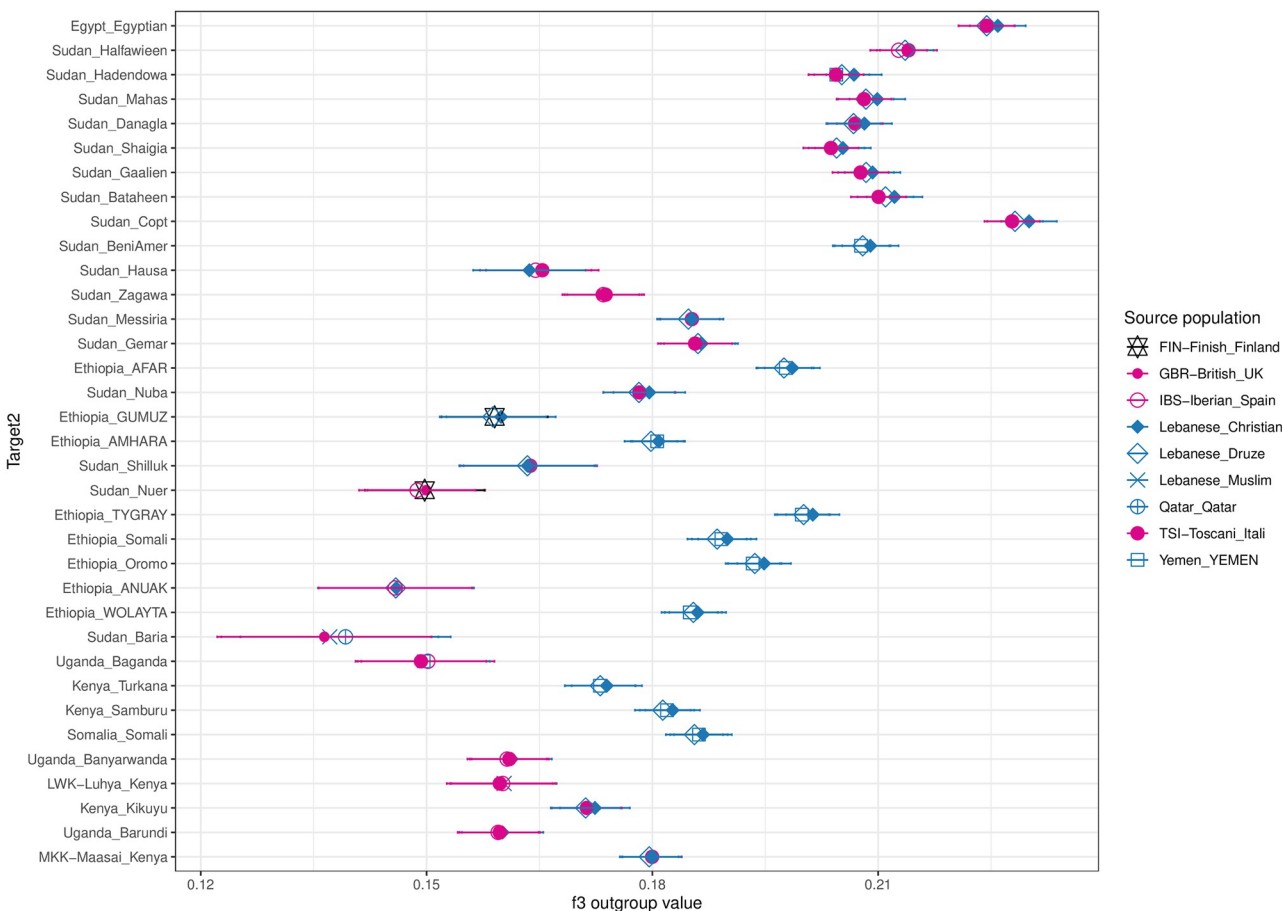

**Fig 2. f₃ outgroup results of the target populations.** Only the top three hits are shown. The f₃ outgroup was calculated for the Eurasian-like ancestry for each target population in the following manner: Target | Source | Ju'hoansi.

then varied the Eurasian sources. We tried both providing one Eurasian population at a time as well as providing two Eurasian populations at the same time. These scenarios were repeated under a 2-way, 3-way, and 4-way admixture scenario, that is using two, three, and four ancestral sources with the four or five reference populations respectively. All 35 target populations were investigated under these differing permutations. Since evaluating the best model can be non-trivial and require lots of manual curation we opted to use MOSAIC's $R^2$ metric (genomic fit) to evaluate the best model. In general, the simpler models performed better, all of the 2-way scenarios outperformed their equivalent (using the same populations) 3-way and 4-way admixture scenarios. Though using two Eurasian populations as sources outperformed a single source. This could be because our dataset does not contain a good match to the original source. These $R^2$ values can be found in S1 Table. This dataset is referred to as the "Best by $R^2$" or simply "$R^2$" dataset in the rest of the study. These runs thus produce two inferred ancestral sources, one Non-African and one African. Five of the target populations did not generate a Non-African source as the closest fit determined by Fst, these were LWK-Luhya_Kenya, Sudan_Baria, Sudan_Hausa, Sudan_Nuer, and Uganda_Baganda thus none of these populations are shown in the best by $R^2$ analysis. This second dataset is thus the best dataset that we achieved using a parametric approach. Ancestry tract length distribution plots for both of these datasets were generated and are available for download from DOI:10.17044/scilifelab.23957703.

## Dating Eurasian admixture in Northeast Africa

For both the $f_3$ outgroup-based approach and the $R^2$ approach above, we determined the admixture date (in generations) from MOSAIC's co-ancestry curves for most Eurasian-like constructed ancestry. The results of this dating can be seen in Fig 3 with the best source based on $f_3$ in A, and D and the dates based on the runs with the highest $R^2$ value in B and E.

Given a generation time of 29 years, this gives a time span ranging from 72.5 years ago for the Nilotic-speaking Anuak to 1027 years ago for the Cushitic-speaking Afar, both from Ethiopia for the best by $R^2$ dataset [40]. In the best by $f_3$ dataset the range is smaller ranging from 84 years for the Eastern Sudanic-speaking Nuer (South Sudan) to 940 for the Semitic-speaking Bataheen (Sudan).

To visualize the correlation between linguistic classification and the inferred admixture date we generated dot plots of the dates per linguistic group as well as the larger linguistic families, see Fig 4.

We compared these dates to the categorical information we had about the populations, that is Country, Linguistic group (e.g. Semitic), or larger Linguistic family (e.g. Afro-Asiatic) using a two-way ANOVA, S4 Table. We find that only larger linguistic families correlated significantly with the detected admixture dates for the best by $f_3$ dataset, S4A Table. The same pattern

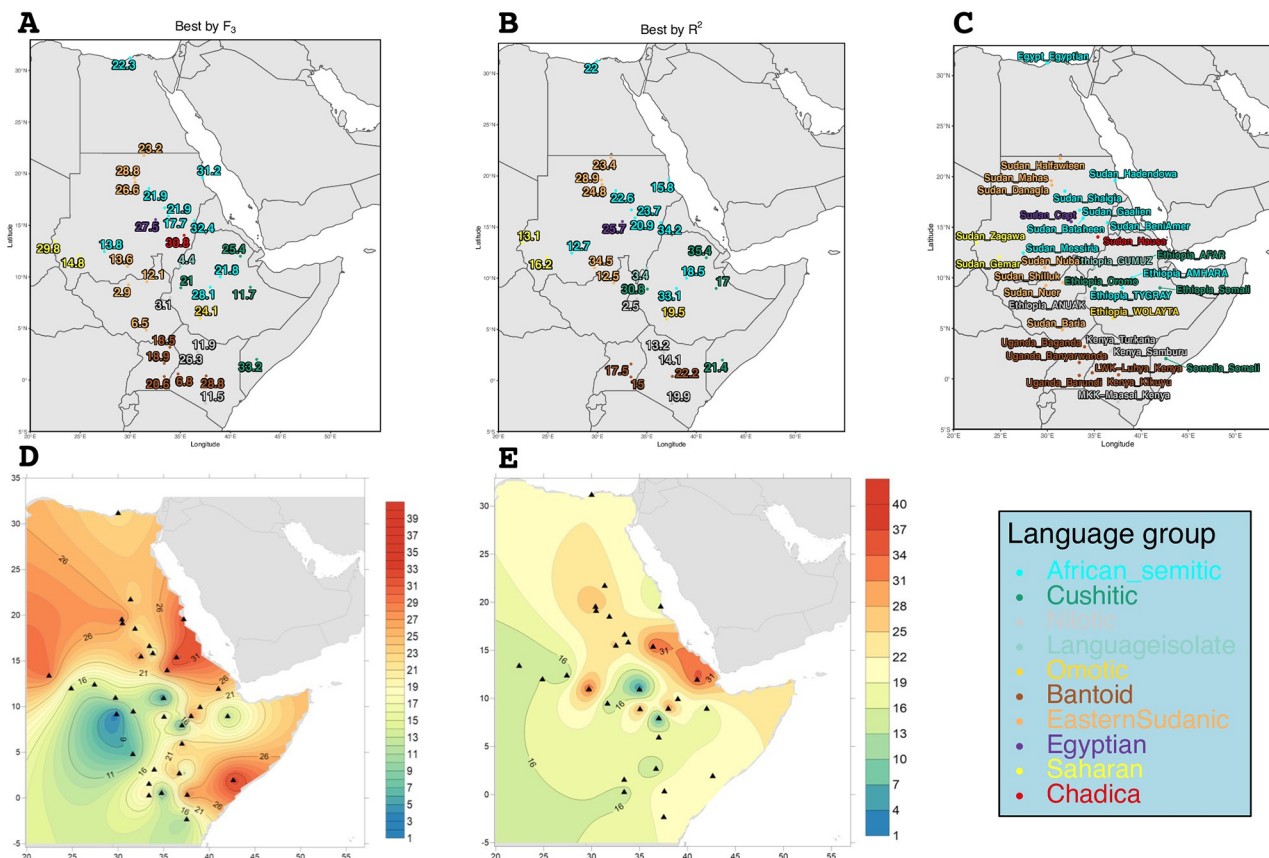

**Fig 3. Admixture dating in generation for the Eurasian-like ancestry from MOSAIC.** A and D contain data for the best source as determined by $f_3$ whilst B and E illustrate the dataset determined by the best on $R^2$ value. A and B are the admixture date in generations, C is the target population locations, and D and E are the same data but plotted over the study area surface using Kriging interpolation. The numbers here represent the major breaks (black lines). Note that some populations did not find a Eurasian source in the best by $R^2$ runs and thus do not have a date.

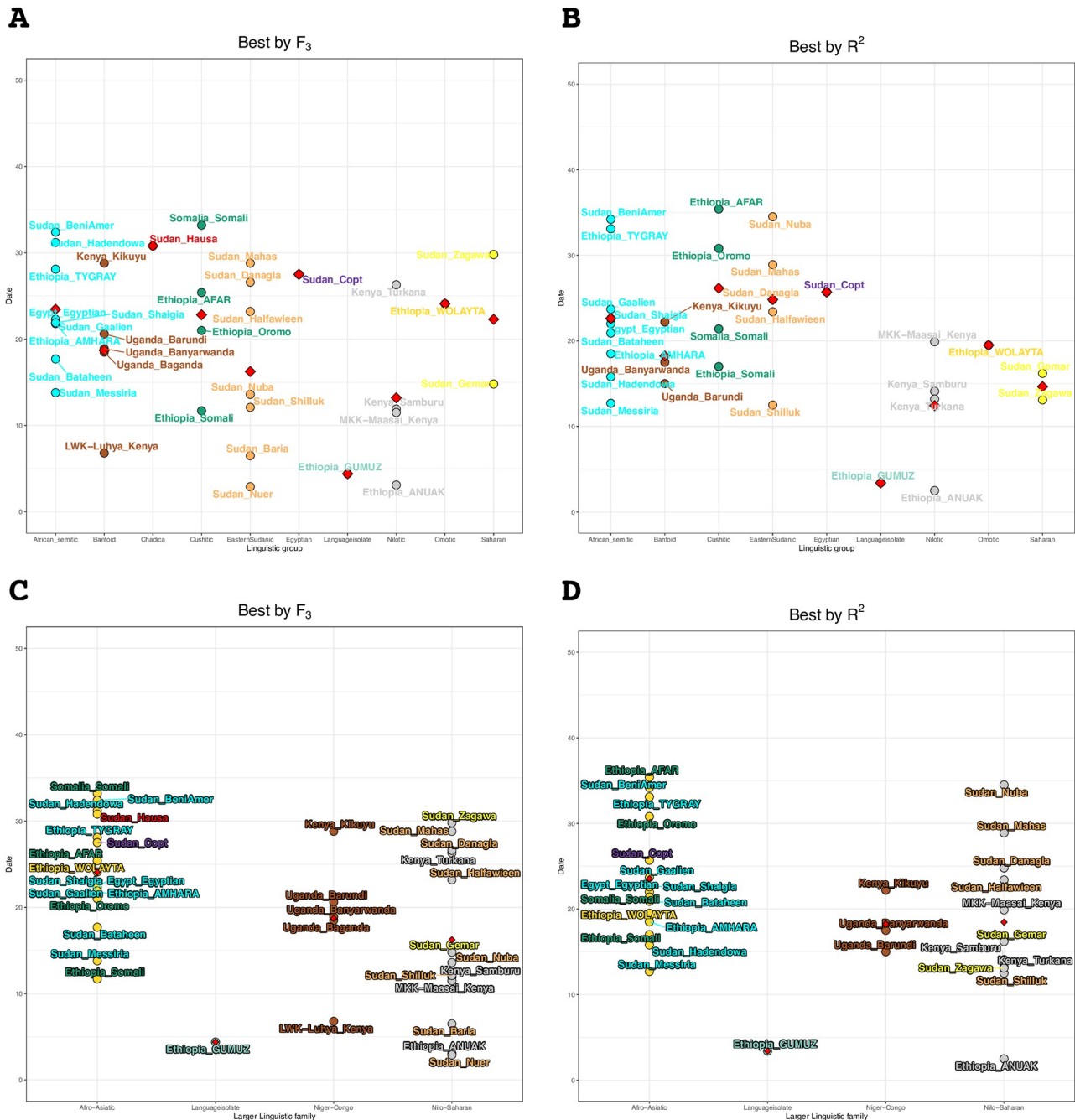

**Fig 4. Dot plot representations of the admixture dating in generation for the most Eurasian-like ancestry from MOSAIC.** A and C contain data for the best source as determined by $f_3$ whilst B and D illustrate the dataset determined by the best on $R^2$ value. A and B are per smaller linguistic classifications whilst C and D show the same data but are divided into linguistic families. The red triangle represents the mean value.

where the lowest p-value is observed for the larger linguistic family is true also for the $R^2$ dataset but without reaching significance, S4B Table. We also test whether there was a spatial correlation to the admixture dates. This was done by comparing the great circle distance between Tel Aviv (a coastal location in the Levant) as well as Sanaa (the capital of Yemen) and each population's sampling location, S21 Fig. For the $R^2$ values by each linguistic family see S3

Table. For the distance from Tel Aviv, we find a low but significant correlation for both data-sets, $R_2$ of 0.088 for the best by $f_3$ dataset and 0.067 for the best by $R^2$ data. We find weaker, but significant, support for the distance from Sanaa in both datasets $R^2$ 0.048 (p = 0.001) for the best by $f_3$ dataset and $R_2$ 0.031 (p = 0.002), S21 Fig.

As there were some discrepancies between the two dating approaches we compared the dates to each other by plotting the dates from the $f_3$ dataset against the $R^2$ dataset, see Fig 5.

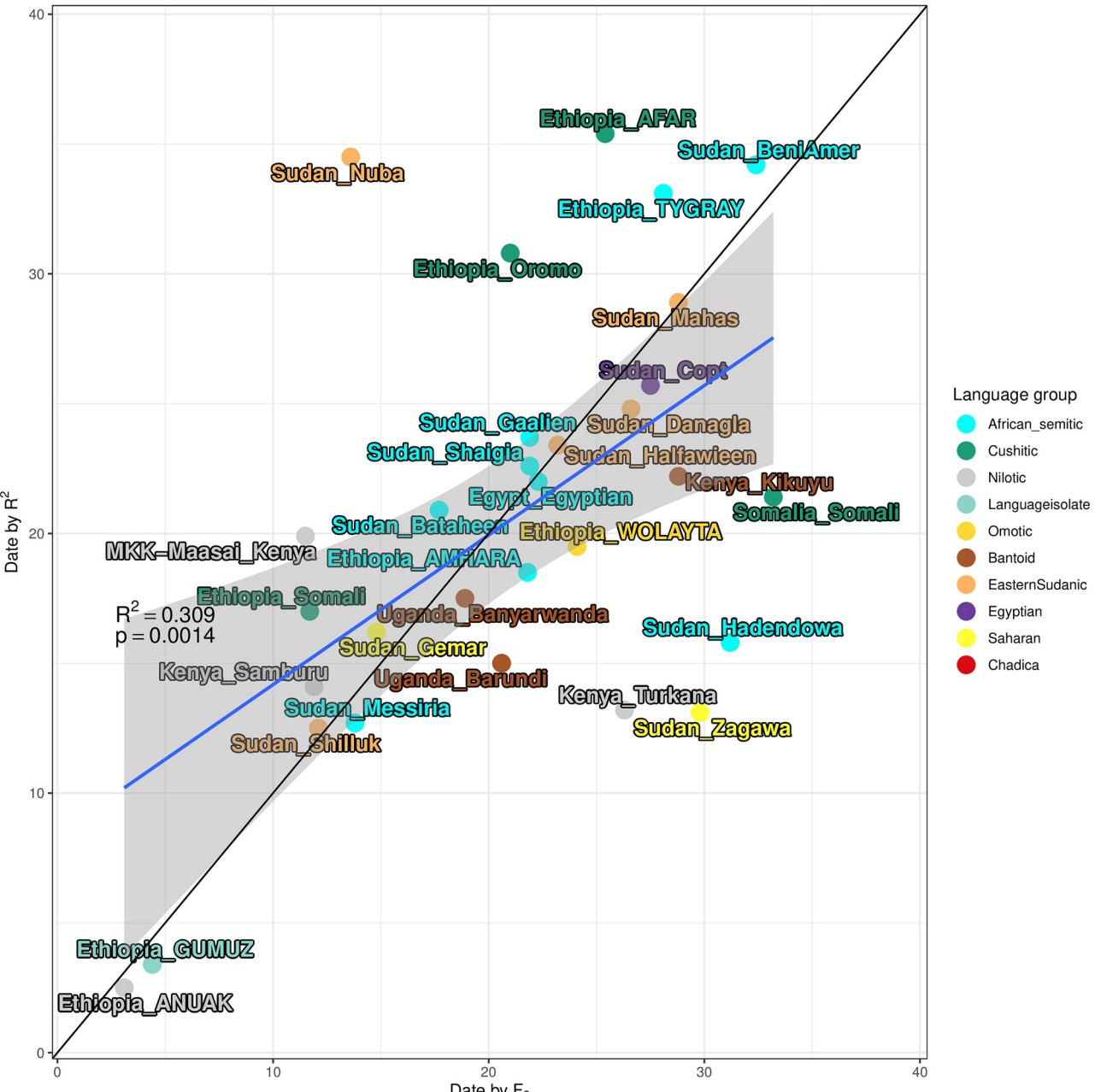

**Fig 5. Correlation between dates from the two approaches.** Linear regression (blue line) comparing the admixture dates of the Eurasian-like ancestry from the best by $f_3$ to the best by $R^2$ dataset. The grey area represents the 95% confidence interval. The black line is X = Y, i.e. same date in both approaches.

Then we performed linear regression on the data. This resulted in a correlation ($R^2$) of 0.3093 with a p-value of 0.001417. The majority of target populations fall within the 95% confidence interval, the gray area in Fig 5. Notable exceptions are the Sudanese Nuba and the Ethiopian Afar populations that have much older dates in the $R^2$ scenario, and the Sudanese Zagawa and Hadendowa, who display the opposite pattern with much older dates in the best by $f_3$ scenario. Most other populations that deviate from either line do so with only a few generations. From our analyses, there is nothing in particular that makes these four populations stand out from nearby populations such as in the ADMIXTURE or PCA. The MOSAIC metric such as $R^2$, $R_{st}$ etc is also in line with comparable populations. The Sudanese Nuba are known to be a relatively heterogeneous group [41], but that is not reflected in our analyses of population structure, see Fig 1 and S6 Fig.

We also compared our inferred dates to LD-based Malder [42] for both datasets S23 and S24 Figs. Malder generally only inferred a single admixture event, thus making the interpretation of a comparison between the two methods somewhat difficult.

## Discussion

In this study, we investigate the patterns of different genetic ancestries in Northeast African populations, focusing on Eurasian back migrations. We inferred population structure using both global and local ancestry methods. Using the local ancestry method MOSAIC we identify regions of Northeast African populations' genomes with Eurasian ancestry. We also attempt to date this admixture. In our approach, we start from the modern-day groups and try to infer patterns of past interactions by analyzing their genomes. We however acknowledge that Northeast Africa is a region with a complex history spanning thousands of years. The expansions and contractions, rise and fall of states, kingdoms, and empires across the region have had a major impact on the formation, dissolution, and current distributions of the sampled communities in this study. We, therefore, recognize that the groups included in this study are modern-day populations that were created by introgression/interaction/assimilation events in the past and should not be seen as unchanged entities that represent exact past distributions of groups. For example, the interpretation of the dating for Nilotic speakers from East Africa, the Maasai, Turkana, and Samburu is complex since they only relatively recently reached their current-day distributions through expansions from Sudan and Uganda within the last few centuries [43].

Dating the admixture of different groups with each other is of great interest to population geneticists. Having a date for when the mixing of two groups occurred allows us to incorporate other types of independent evidence into our analyses, such as written or oral history or linguistic information, thus a big part of the effort in this study and discussion is spent on our inferred dates of admixture. The "best by $f_3$" analysis is our attempt to propose a scenario that best fits the previously known genetic history of the region, whilst the "best by $R^2$" analysis, based on the genomic fit ($R^2$), is intended as a less constrained scenario for picking out Eurasian ancestry in Northeast African groups. As shown in Fig 5 the two approaches result in similar dates for most of the populations.

Population structure inferences illustrate the complex genetic history of Northeast African populations. Larger patterns of genetic associations between many of the world's distinct human lineages are reflected in Northeast African genomes. The hunter-gatherer's ancestry highlights the deep history of the region and its people and that this ancestry remains within the East African populations. The southern part of the region has a closer genetic affinity to West African groups, a result of the Bantu expansion and several of these populations also speak Bantu languages today. That the Bantu expansion did not continue further into the

region could be a result of geographical barriers such as the Ethiopian Highlands and the dry regions of the Horn of Africa, indicated by our FEEMS analysis in Fig 1D or as suggested by [15] that the Northeast African Nilotic speaking herders (such as the Dinka and Nuer), who have remained relatively isolated from other groups, could have formed a buffer against the Bantu expansion continuing further into Northeast Africa.

Eurasian admixture has had a large influence on the genomes of Northeast African groups. The Egyptian and Sudanese Copt populations for instance are genetically very similar to Middle Eastern groups rather than to other African populations. The pattern is true also for the rest of North Africa and present as early as at least 15 000 years ago [10] though not investigated here. The Copts look genetically similar to the Egyptians from Cairo, see Fig 1A and 1C, this is not unsurprising given that the Copts arrived in Sudan around 200 years ago from Egypt and seem to have lived relatively isolated since then [15]. Our admixture date for the Copts (with Eurasians) was inferred to be 27.5 for the $f_3$ analysis and 25.7 for the $R^2$ and around 22 generations for the Egyptians. Thus this admixture took place around the 14th century.

Further south in the region, we continue to see the impact of past Eurasian admixture. Northeast African populations from Sudan and Ethiopia positions' in the PCA plots are being drawn towards Eurasian populations, S6 Fig. ADMIXTURE analyses recapitulate this pattern where Northeast African groups share the component maximized in Middle Eastern groups (pink component at K = 6, Fig S2 Fig). The Sudanese data in our study is mainly from [15] who also investigated the time and sources of admixture in Sudanese populations. [15] investigated a simpler admixture scenario with only two putative sources, namely the Sudanese Nuer and the TSI (Tuscan) to represent the admixture of a Sudanese basal population with a Eurasian source. This is most similar to our $R^2$ approach in which we picked the scenario with the best genomic fit ($R^2$) and for two Eurasian sources in each run and then picked the two Eurasian sources that produced the best genomic fit ($R^2$ value). Our findings are generally in agreement, particularly for the Eurasian admixture dates that is the primary focus of our study.

In the area of current-day Sudan and South Sudan, there is a clear divide between the Eastern Sudanic- and Semitic-speaking groups from Sudan, and the South Sudanese groups, as well as the Saharan-speaking Sudanese groups. This divide can be seen both with regards to global ancestry as well as their inferred admixture dates for their Eurasian ancestries. Dongola had been the capital of the Nubian Kingdom and the fall of Dongola in 1317 to Mameluke forces meant the start of Arab and Islamic dominance south of the borders of Egypt. Many of the Semitic speakers in our dataset have their Eurasian admixture dated to this time—around 20 generations ago. The exception is mainly the Southern Semitic speakers such as the Beni-Amer and Tygray whose dates are slightly older at around 30 generations ago. Around 30 generations ago is also the inferred date for the Ethiopian Cuschitic-speaking Afar and Oromo (though Oromo had a generation time of 21 for the best by $f_3$). South Sudanese groups however stayed largely isolated, this pattern is evident in the ADMIXTURE analysis, as the populations around South Sudan are represented by a specific component (the black component at K = 5 and onward) with very little of the non-African (pink) component that we find in most other North-East African groups, indicating their isolation and genetic homogeneity compared to other populations.

Previous studies that investigated Ethiopian population structure, observed clustering based on linguistic families [4, 14, 15]. This pattern is recapitulated in our analysis, both from the population inference methods as well as the admixture dating. The Omotic-speaking Ari populations form their own small cluster (PC 1 vs PC 3 in S6 Fig), a reflection of their segregated status within Ethiopia [4]. The Gumuz (Language isolate) and Anuak (Nilotic) display very little Eurasian admixture, and given that the date that we infer is only a few generations

ago, it could be that they received this Eurasian admixture through secondary admixture with another neighboring group a few generations ago, or that it's an effect of recent or ongoing admixture.

Within the northeast African geographic space, the analysis using FEEMS recapitulates expected natural barriers to migration such as the Red Sea, the Gulf of Aden, the Persian Gulf, and the Sahara Desert. In addition to clear geographical barriers, we also see evidence of linguistic and cultural barriers. One obvious example is the low migration rate between the Ethiopian Somali and the other Ethiopian populations, and as expected high migration rate is inferred between the different Somali groups. The Great Rift Valley forms a natural barrier across Ethiopia with highlands on both sides of the rift. A previous study looking at Ethiopian genetics found a significant association of genetic similarity to elevation, ethnicity, and first language, and interestingly not a second language nor religion [13].

Along the Red Sea coast of Eritrea and Sudan, we find a region of high gene flow extending into northern Ethiopia and into the Great Rift Valley, Fig 1D. This region corresponds well to the pink component in Fig 1A and 1B which seems to represent Yemeni ancestry. f3 visualizations also indicates higher geneflow from Arabian groups in this area relative to more northern and southern latitudes, Fig 2. It is also a region in which we infer some of the oldest inferred admixture dates. These observations, as well as the shared linguistics of South Semitic (as South Semitic languages that are found in Yemen, Oman, Eritrea, and Ethiopia [44]), indicate a close connection between Eritrea, and Ethiopia to the south of the Arabian peninsula and present-day Yemen. The Kingdom of Aksum (or the Aksumite Empire) encompassing Eastern Sudan, Northern Ethiopia, Eritrea, Djibouti and across the Red Sea into Yemen, thrived between the 1:th and 7:th century AD, as trade along the Red Sea increased and the trade along the Nile decreased. Both Rome and Byzantium traded with the Indian Subcontinent and artifacts from these Kingdoms can be found at Aksumite sites, [45, 46]. The Semitic-speaking Ethiopian populations also group together with the Middle Eastern populations in the UMAP analysis, S20 Fig. These admixture events could come as the result of the Red Sea trade. Aksum collapsed in the 8th century as Islam started to expand and control over the Red Sea trade shifted to the Near East [47].

Previous ancestry deconvolution studies pointed at Levantine sources for the Eurasian admixture in Northeast Africans rather than Arabic groups [4, 6]. We find that the pattern is more complex with different source populations in different regions, see Fig 1B and 1C as well as Fig 2. Levantine contributions are seen more towards the north and contributions from Arabian peninsula groups are seen more at lower latitudes, Figs 1B, 1C and 2.

The fact that both approaches for admixture dating produced populations from the same country that had the most extreme difference in Eurasian admixture dating, highlights the heterogeneous nature of North-East African genetics and how little explanatory power country borders have on population structure. It is, not unexpectedly so, rather geographic, linguistic, and cultural borders that explain the degree of genetic interconnections between groups.

The major linguistic family was the only factor that was significant (and only for the best by $f_3$) in our ANOVA test of the available categories, S4 Table. The linear regression analysis of distance from the Levant, S21A and S21B Fig, also produced a significant fit with a negative coefficient indicating more recent admixture dates further from the Levant—this is likely driven by the younger dates for the populations in and around South Sudan. The same pattern was observed when comparing the distance to Sanaa, albeit with a smaller slope of the line and larger p-values (S21C and S21D Fig).

One possible explanation for this phenomenon could be that populations with little or no previous Eurasian admixture would have their inferred admixture date affected more by recent

Eurasian admixture than populations that experienced larger admixture in the past. In other words, most, if not all, of the populations in this study have or have had admixture with populations from the Middle East during the Arab expansion, and this newer admixture is obscuring older admixture patterns. The groups with younger inferred dates in our analysis thus likely have less older admixture contributions.

Our study thus points to that the distribution of Eurasian-like ancestry in Eastern and North-Eastern African populations is mostly an effect of more recent migrations (many of them recorded in historical texts) rather than ancient events related to the advent of pastoralism in the region at large, as indicated by ancient DNA studies [5]). Identifying the impact of ancient events on populations was not feasible when the original pattern has been distorted or masked by subsequent admixture events. To fully explore the question of Eurasian admixture into Africa over larger timescales likely requires population-level aDNA, especially of the early East African hunter-gatherers such as Mota, and the various in-moving groups, including those containing Eurasian admixture.

## Conclusions

North-Eastern Africa is a vast region with complex histories of migrations and admixture. It was not possible to identify one source or origin of Eurasian admixture in the region, rather different populations have experienced admixture at different times, at varying degrees, and from different external sources. Although slight trends were observed linked to language grouping and geography, the overall pattern proved to be complex and specific to certain population groups. Previous studies have highlighted these events in distinct regions or countries in Northern and Eastern Africa, whilst we in this study have tried to combine them with a specific emphasis on the Eurasian admixture in modern-day populations.

## Supporting information

**S1 Table. Top two Eurasian source populations identified by their haplotype fit to the genomes ($R^2$ value from MOSAIC) for each target population.**
(PDF)

**S2 Table. $f_3$ outgroup result grouped by language family of the target populations, top 5 hits show.** The $f_3$ outgroup was calculated for the most Eurasian-like ancestry for each target population in the following manner: Target | Source | Ju|'hoansi.
(PDF)

**S3 Table. $R^2$ values of linear regression between admixture date and distance from Tel Aviv in the top row by each linguistic group.** The Best by $f_3$ dataset is in the leftmost columns while the Best by $R^2$ dataset is on the right. Some linguistic groups have only one target population so no value whilst Saharan two populations which yields a $R^2$ of 1.
(PDF)

**S4 Table. Two way ANOVA statistics for the two different models.** Comparisons are between the Admixture date with the factors Country, Linguistic group and Larger linguistic family (Meta.Lang.Group). The Asterisk indicates significant values. In A it's the admixture date obtained from the best source determined by $f_3$ outgroup and in B it's the dates of admixture for the source with the highest.
(PDF)

**S5 Table. Missingness by population for the Ancestry deconvolution.** Min and max is the individual lowest and highest missingness for that population. Since non-Eurasian regions of

the target's genomes were set to missing, this measure is the inverse of the amount of Eurasian ancestry inferred for each individual and population in the best by $f_3$ dataset.
(PDF)

**S6 Table. Overview of populations used in the study and their linguistic information.**
(XLSX)

**S1 Fig. Location for populations included in this study.** Colours indicate linguistic groups. Made with Natural Earth.
(PDF)

**S2 Fig. PONG visualization of 15 K's of unsupervised ADMIXTURE analysis.** 50 iterations for the full dataset. The best identified K through cross-validation was K = 13.
(PDF)

**S3 Fig. Average cross-validation (CV) error for the 50 repetitions.** The K with the lowest CV error was K = 13, indicated by the horizontal line.
(PDF)

**S4 Fig. North-East African target populations used in the study.** Labels are by country and colouring by linguistic family. African_Semitic was used just more easy to distinguish between the investigated populations (target) and the Middle Eastern Semitic populations. Made with Natural Earth.
(PDF)

**S5 Fig. Principal component analysis with each value in the PCA plots, is the projection of the data on the eigenvectors, scaled by the eigenvalues.** Values within parenthesis are the PC loading. Populations are coloured by linguistic group.
(PDF)

**S6 Fig. Principal component analysis.**
(PDF)

**S7 Fig. Principal component analysis.**
(PDF)

**S8 Fig. Principal component analysis.**
(PDF)

**S9 Fig. Principal component analysis.**
(PDF)

**S10 Fig. Principal component analysis.**
(PDF)

**S11 Fig. Principal component analysis.**
(PDF)

**S12 Fig. Principal component analysis.**
(PDF)

**S13 Fig. Principal component analysis.**
(PDF)

**S14 Fig. Principal component analysis.**
(PDF)

**S15 Fig. Principal component analysis.**
(PDF)

**S16 Fig. Principal component analysis.**
(PDF)

**S17 Fig. Principal component analysis.**
(PDF)

**S18 Fig. Principal component analysis.**
(PDF)

**S19 Fig. Principal component analysis.**
(PDF)

**S20 Fig. Uniform Manifold Approximation and Projection for Dimension Reduction on the full dataset.** Colours are the same as in S1 Fig.
(PDF)

**S21 Fig. Linear regression comparing the great circle distance in kilometers from Tel Aviv and Sanaa compared to admixture date of the Eurasian ancestry estimations.** The blue line is the fitted linear regression line and the grey area represents the 95% confidence interval of the standard error. A) Distance from Tel Aviv for the best by $f_3$ dataset. B) Distance from Tel Aviv for the best by $R^2$ dataset. C) Distance from Sanaa for the best by $f_3$ dataset. D) Distance from Sanaa for the best by $R^2$.
(PDF)

**S22 Fig. f4 test comparing Lebanese to Yemenite ancestry for each of the target populations.**
(PDF)

**S23 Fig. MALDER vs MOSAIC dates.** For the best by $f_3$ dataset, using the same source populations as in the corresponding MOSAIC analysis. Only populations that Malder estimated had one event are shown. The populations for which Malder inferred two admixture events were: Egypt_Egyptia 40 and 6 generations ago, Sudan_Halfawieen 87 and 7 generations ago, and Sudan_Mahas 94 and 12 generations ago. The blue line is the fitted linear regression line and the grey area represents the 95% confidence interval of the standard error.
(PDF)

**S24 Fig. MALDER vs MOSAIC dates.** For the best by $R^2$ dataset, using the same source populations as in the corresponding MOSAIC analysis. Only populations that Malder estimated had one event are shown. The populations for which Malder inferred two admixture events were: Egypt_Egyptian 39 and 8 generations ago, Kenya_Turkana 8 and 164 generations ago, Sudan_Halfawieen 77 and 6 generations ago, and Sudan_Mahas 81 and 12 generations ago. The blue line is the fitted linear regression line and the grey area represents the 95% confidence interval of the standard error.
(PDF)

## Acknowledgments

The computational analyses were done through the Swedish National Infrastructure for Computing (SNIC) at Uppmax. Authorized NIH Data Access Committee (DAC) granted data access to Carina Schlebusch for the controlled-access genetic data analyzed in this study that were previously deposited by Scheinfeldt et al. 2019 in the NIH dbGAP repository (accession

code phs001780.v1.p1; date of approval: 2019-05-17). For the genome-wide genotype data from the Patin et al. 2017 study (EGA accessory number EGAD00010001209), data access was granted via European GenomePhenome Archive (EGA) by the GEH Data Access Committee EGAC00001000139. A special thanks to the author of MOSAIC Michael Salter-Townshend for the discussion on how the best perform the ancestry deconvolution, to Carolina Bernhardsson for help with plotting, and Cesar Fortes-Lima for help with the study design.

## Author Contributions

**Conceptualization:** Rickard Hammarén, Carina M. Schlebusch.

**Data curation:** Rickard Hammarén, Carina M. Schlebusch.

**Formal analysis:** Rickard Hammarén.

**Funding acquisition:** Carina M. Schlebusch.

**Investigation:** Rickard Hammarén, Steven T. Goldstein, Carina M. Schlebusch.

**Methodology:** Rickard Hammarén, Carina M. Schlebusch.

**Project administration:** Carina M. Schlebusch.

**Resources:** Rickard Hammarén, Carina M. Schlebusch.

**Software:** Rickard Hammarén.

**Supervision:** Carina M. Schlebusch.

**Validation:** Rickard Hammarén.

**Visualization:** Rickard Hammarén.

**Writing – original draft:** Rickard Hammarén, Carina M. Schlebusch.

**Writing – review & editing:** Rickard Hammarén, Steven T. Goldstein, Carina M. Schlebusch.

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
