## [Decision Letter · Decision Letter 0]

15 Nov 2022

PONE-D-22-25736Eurasian back-migrations into Northeast Africa was a complex and multifaceted processPLOS ONE

Dear Dr. Schlebusch,

Thank you for submitting your manuscript to PLOS ONE. After careful consideration, we feel that it has merit but does not fully meet PLOS ONE’s publication criteria as it currently stands. Therefore, we invite you to submit a revised version of the manuscript that addresses the points raised during the review process.

We look forward to receiving your revised manuscript.

Kind regards,

Francesc Calafell

Academic Editor

PLOS ONE

Journal Requirements:

   "The computation and data handling were enabled by resources provided by the Swedish National Infrastructure for Computing (SNIC) at Uppmax partially funded by the Swedish Research Council through grantagreement no. 2018-05973. Authorized NIH Data Access Committee (DAC) granted data access to Carina

Schlebusch for the controlled-access genetic data analysed in this study that were previously deposited byScheinfeldt et al. 2019 in the NIH dbGAP repository (accession code phs001780.v1.p1; date of approval:2019-05-17). For the genome-wide genotype data from the Patin et al. 2017 study (EGA accessory numberEGAD00010001209), data access was granted via European GenomePhenome Archive (EGA) by the GEH

Data Access Committee EGAC00001000139. This project was supported by funding to CS from the European Research Council (ERC) under the European Union’s Horizon 2020 research and innovation programme(grant agreement No. 759933). A special thanks to the author of MOSAIC Michael Salter-Townshend for

discussion on how the best perform the ancestry deconvolution, to Carolina Bernhardsson for help with plotting and Cesar Fortes-Lima for help with the study design."

 "No: The funders had no role in study design, data collection and analysis, decision to publish, or preparation of the manuscript."

5. Please ensure that you include a title page within your main document. You should list all authors and all affiliations as per our author instructions and clearly indicate the corresponding author.

6. We note that Supplementary Figure 1 in your submission contain [map/satellite] images which may be copyrighted. All PLOS content is published under the Creative Commons Attribution License (CC BY 4.0), which means that the manuscript, images, and Supporting Information files will be freely available online, and any third party is permitted to access, download, copy, distribute, and use these materials in any way, even commercially, with proper attribution. For these reasons, we cannot publish previously copyrighted maps or satellite images created using proprietary data, such as Google software (Google Maps, Street View, and Earth). For more information, see our copyright guidelines: http://journals.plos.org/plosone/s/licenses-and-copyright.

a. You may seek permission from the original copyright holder of Supplementary Figure 1 to publish the content specifically under the CC BY 4.0 license.  

Additional Editor Comments:

Please take into account the clarifications and suggestions for improvement that the reviewers contributed, and in particular try to address the methodological concerns expressed by reviewer #3.

Reviewers' comments:

Reviewer's Responses to Questions

**Comments to the Author**

1. Is the manuscript technically sound, and do the data support the conclusions?

Reviewer #1: Yes

Reviewer #2: Yes

Reviewer #3: Partly

2. Has the statistical analysis been performed appropriately and rigorously? 

Reviewer #1: Yes

Reviewer #2: Yes

Reviewer #3: Yes

3. Have the authors made all data underlying the findings in their manuscript fully available?

Reviewer #1: Yes

Reviewer #2: Yes

Reviewer #3: Yes

4. Is the manuscript presented in an intelligible fashion and written in standard English?

Reviewer #1: Yes

Reviewer #2: Yes

Reviewer #3: Yes

5. Review Comments to the Author

Reviewer #1: The manuscript by Hammaren and colleagues addresses an interesting question about North African populations: the origin of the Eurasian back-migrations. I highly appreciate that the authors reanalyzed data from previous papers that could be further explored. However, the answer to the main question is not clearly discussed. The paper would improve with a clearer discussion.

In general, there is too much emphasis and sometimes overinterpretation of the ADMIXTURE analyses. The discussion in mainly driven by those results, but the most novel and interesting part are the f3 analyses after local ancestry:

- The text is too descriptive and difficult to follow. Some of the descriptions about ADMIXTURE components and PC could be shortened.

- You might need to review the colors linked to the ADMIXTURE plots, sometimes they do not match with the description in the text:

-“Light green component that emerges is K=7 it is maximized in the Sabue”. This color is maximed in Namibia_Nama SouthAfrica_Juhoansi.

-“blush pink component highest in the Qatari”. It is higher in Yemen.

-“Gumuz and Sabue have a high proportion of the dark green component.” Is this the black component?

-“blue component maximized in Middle Eastern groups at K=7, Figure 2.” I think this is figure 1.

- Also, you might want to check the labels of the ADMIXTURE plots, the names are sometimes duplicated and not clear.

- I would suggest providing a visualization of the f3 results in Table 1. And discussing these results and their interpretations in the discussion.

- Regarding ADMIXTURE interpretations, one example is the hunter-gatherer ancestry, which is mentioned through the text at different points. Are you claiming that it corresponds to one of their ancestral components in ADMIXTURE? It is too speculative without ancient data. Also, talking about “ancestries” from ADMIXTURE analyses is in most cases speculative, I would suggest checking these interpretations and the language used.

- In the discussion, the paragraph “Previous recent ancestry deconvolution studies pointed at Levantine”. You are repeating the description results that were already mentioned, but it is not clear what is the interpretation.

About the methodology:

- It is not explained how you treat the data after the local ancestry. Do you mask the rest of the genomes setting it as missing? If that is the case, which proportion of the genome is masked and therefore what are the missingness values?

- Could the use of CEU for the local ancestry bias the following analyses about the specific source for this ancestry? It is good that the analyses are repeated with the best source from f3 afterwards, but I wonder if this could produce a circular result (i.e. if the first analyses using CEU are biasing the f3 estimates, so the next analyses would be also biased). Could you comment it further?

- You mention: “Though using two Eurasian populations as sources outperformed a single source.” Is it possible that this happens because you do not have the best source and then it is more likely that the source is coming from two different ones?

- Is it possible to estimate confidence intervals for the admixture dating? Considering the differences observed between the “f3” and “R2” estimations it would be important. The correlation between both admixture dates is not strongly significant and there are some outliers. It is briefly disused in the discussion, but why specifically do you think these differences are found? For instance, is there a worst correlation among those with more Eurasian ancestry?

- To see if the masking is necessary, I suggest using an f4 test comparing between possible sources of the Eurasian ancestry in the different Northeast African populations, i.e. f4(Levant, Arab; Northeast Africa, Ju’hoansi).

- It would be interesting to see the ancestry proportions estimated with MOSAIC. For instance, comparing them with the admixture dates could bring more information about the different demographic processes.

- What are the sources of the admixture dating, is it always the Eurasian population and an African population? For instance, in the discussion, when talking about the Copts and Egyptians, it is not clear what are the sources of the admixture dates described.

- The discussion about how more recent admixture events could be masking the older ones is not supported with specific analyses. Maybe the authors could cite at least some other works.

- There are some other places where a reference would help, for instance, “The pattern is true also for the rest of North Africa, though not investigated here.”

On a minor note, English is readable, but needs to be checked. There are some repetitions of words and gramma mistakes. I am not listing them because there are many, so please check it carefully. Some that I found:

-“are evident across the region of interest interest we used the FEEMS”

-The title in the results: “Dating Eurasian admixture dating in Northeast Africa.” You do not need the second “dating”.

Reviewer #2: In the manuscript by Hammaren et al the authors attempt to disentangle some of the complexity in the genetic history of present day populations from North East Africa. They begin by collecting a large dataset of publicly available present-day data spanning all of Africa, the Near East and Europe, and then apply a variety of methods in attempts to identify present-day populations that appear distinctively similar to the introgressed Eurasian fragments in North East African groups and could thus be good proxies, or at least shed light onto the past dynamics of the region.

Although their final conclusion is that their attempt is not as successful as they would like, it does take time to get there, and I think this is something that could be made clearer earlier on, although I also grant it might be a different in writing style. But personally, I found the concluding remarks somewhat undermined the rest of the discussion, for all that I very much agree with the key points made in that final section, and I wonder if it might be more effective if brought in earlier. Beyond that, I have mostly minor comments, and no concerns to raise about the scientific content of the manuscript as much as about its presentation.

1. Title: should be either "Eurasian back-migrations into Northeast Africa WERE a complex and multifaceted process" or "Eurasian back-migration into Northeast Africa was a complex and multifaceted process"; the current version doesn't have subject/tense agreement.

2. Methods, "4) individuals with at least 10% missingness was removed (plink --mind 0.01)" That is actually a filter for 1% missingness - please confirm what was actually done!

3. Results, "Note that some populations are represented multiple times from diﬀerent original publications, resulting in a total of 97 unique populations." As a sanity check for the dataset merging, do results for the same population across different publications generally agree?

4. Results, "The output from Admixture shown in Figure 1 (for full analysis see Supplementary Figure 2)" It would be good if the figure legend (or somewhere in the main text) made clear that this is a truncated version of Supplementary Figure 2, and the K=13 and 14 panels don't include all clusters (East Asia is missing), as the 'full analysis' is a bit ambiguous here, since not all values of K are shown in the main text fig. Likewise, an explanation of what 'w' is in panel 1D would be helpful for readers not very familiar with FEEMS.

5. Results, "Ancestry tract length distribution plots for both of these datasets was generated and are available on request." If this is a matter of just plots, please add them as supplements to begin with. It'll be easier for everyone than having to dig them up when a reader asks for them in three years!

6. Discussion, "Admixture analyses recapitulate this pattern where Northeast African groups share the blue component maximized in Middle Eastern groups at K=7, Figure 2." That should surely be figure 1?

7. Discussion, "Dongola had been the capital of the Nubian Kingdom and the fall of Dongola in 317 to Mameluke forces meant the start of Arab and Islamic dominance south of the borders of Egypt. Many of the Semitic speakers in our dataset have their Eurasian admixture dated to this time - around 20+ generations ago.", Is there a typo here, or a number missing? Using the value given in the text of 29 years per generation, 20 generations is 580 years ago, and looking at the corresponding figure no numbers are anywhere near close to the 58 generations needed to get to get to 320.

8. Discussion, "One possible explanation for this phenomenon could be that populations with little or no previous Eurasian admixture would have their inferred admixture date eﬀected more by recent Eurasian admixture than population that experienced larger admixture in the past." Effected should be affected.

Reviewer #3: Review of “Eurasian back-migrations into Northeast Africa was a complex and multifaceted process”

In “Eurasian back-migrations into Northeast Africa was a complex and multifaceted process” Rickard Hammaren and colleagues assembled and analysed a genome-wide dataset composed of XXX individuals from several populations focusing on the recent demographic dynamics of the North-Eastern side of the African continent.

In doing so, they apply different population structure and local ancestry analyses. The main claim the authors make is that “the distribution of the Eurasian-like ancestry in the Eastern an North-Eastern African populations is mostly an effect of more recent migrations rather than ancient events related to the advent of pastoralism in the region at large”.

In my opinion this is an interesting but at the same time controversial statement, which requires a carfeul validation analysis. A series of different local ancestry analysis and admixture dating have repeatedly reported a signal of Eurasian-like ancestry entering into the continent at least 3,000 years ago. Although I appreciate the fact that the complexity of the many layers intersecting since at least 15,00 years ago may introduce many bias in previous researches, I can’t see how the analysis conducted in this paper could not affected by similar issues. Moreover, I have previously attempted to use MOSAIC on simulated data, and in our specific setting, MOSAIC was not able to infer the true ancestral fragments, causing a bias in the admixture dating. I did not conduct any further analysis and therefore I do not have any evidence MOSAIC does something wrong, but I would definitely use some other local ancestry inference methods and admixture dating approaches checking if the results hold.

Minor considerations:

Pag: I would clearly state the number of individuals and populations in the final dataset

Pag 3: it is not clear if you removed individuals with missing exceeding 1% or 10% as the description and the command line are discordant.

Pag 3: I think that the UMAP analysis should be described in a more detailed way, and possibly evaluating the outcome of multiple iterations.

Pag 4: Maybe F3 could be written as f3 ?

Pag 5: Given that this article might be of interest to many scholars, I would specify what TSI and RHGs mean.

Pag 6: The last paragraph explaining the approach for the best R2 and f3 was, in my opinion hard to read, I would consider reformulating it.

Pag 7: I think that the Discussions are too long, containing many paragraphs that would probably be a better fit into the results section.

6. PLOS authors have the option to publish the peer review history of their article (what does this mean?). If published, this will include your full peer review and any attached files.

Reviewer #1: No

Reviewer #2: No

Reviewer #3: No

---

## [Decision Letter · Decision Letter 1]

9 Aug 2023

Eurasian back-migration into Northeast Africa was a complex and multifaceted process

PONE-D-22-25736R1

Dear Dr. Carina M Schlebusch,

We’re pleased to inform you that your manuscript has been judged scientifically suitable for publication and will be formally accepted for publication once it meets all outstanding technical requirements. Please note that one reviewer rightly mentioned the availability of data, which I trust the authors would follow. 

Kind regards,

Gyaneshwer Chaubey

Academic Editor

PLOS ONE

Additional Editor Comments (optional):

Reviewers' comments:

Reviewer's Responses to Questions

**Comments to the Author**

1. If the authors have adequately addressed your comments raised in a previous round of review and you feel that this manuscript is now acceptable for publication, you may indicate that here to bypass the “Comments to the Author” section, enter your conflict of interest statement in the “Confidential to Editor” section, and submit your "Accept" recommendation.

Reviewer #1: All comments have been addressed

Reviewer #2: (No Response)

Reviewer #3: All comments have been addressed

2. Is the manuscript technically sound, and do the data support the conclusions?

Reviewer #1: Yes

Reviewer #2: Yes

Reviewer #3: Yes

3. Has the statistical analysis been performed appropriately and rigorously? 

Reviewer #1: Yes

Reviewer #2: Yes

Reviewer #3: Yes

4. Have the authors made all data underlying the findings in their manuscript fully available?

Reviewer #1: Yes

Reviewer #2: No

Reviewer #3: Yes

5. Is the manuscript presented in an intelligible fashion and written in standard English?

Reviewer #1: Yes

Reviewer #2: Yes

Reviewer #3: Yes

6. Review Comments to the Author

Reviewer #1: I appreciate the efforts of the authors to reply to the questions raised in the revision. The paper has improved its readability and I do not have any major questions.

Just a small possible error:

- In line 97: “publications in the following fashion, 1) directly from the text or s 2) if no coordinates”. Is there something missing in the “s” after “or”?

And a small question:

- From the f3 analysis, there are two patterns 1) some populations do not show affinity towards Europeans, 2) other populations do show similar affinity to Lebanese/Qatar/Yemen or European.

Is this correlated with the observation in the ADMIXTURE analyses?

I wonder if in case 2) admixture might come from two different Eurasian sources. Do you think it is the case, or it is not supported by your MOSAIC results?

Reviewer #2: I thank the authors for their response to my comments. I am mostly satisfied, especially as my comments were minor, but their response to point 5 does not meet the PLOS Data policy, which states that all data should be made available, either as supplements or through deposition in a public repository (such as Figshare etc). I agree with their decision not to add 70 supplementary figures to the manuscript file, but then the figures, which are deemed worth mentioning in the text, should be publicly deposited elsewhere, or mention to them and the relevant ancestry tracts omitted from the manuscript.

Reviewer #3: The authors replied to my main concerns, and I do not have any further consideration.

I would recommend the revsed version for pubblication.

7. PLOS authors have the option to publish the peer review history of their article (what does this mean?). If published, this will include your full peer review and any attached files.

Reviewer #1: No

Reviewer #2: No

Reviewer #3: No

---

## [Editor Report · Acceptance letter]

18 Aug 2023

PONE-D-22-25736R1 

Eurasian back-migration into Northeast Africa was a complex and multifaceted process 

Dear Dr. Schlebusch:

I'm pleased to inform you that your manuscript has been deemed suitable for publication in PLOS ONE. Congratulations! Your manuscript is now with our production department. 

Kind regards, 

on behalf of

Gyaneshwer Chaubey 

Academic Editor

PLOS ONE